# Spatial and Social Behavior of Single and Coupled Individuals of Both Sexes during COVID-19 Lockdown Regime in Russia

**DOI:** 10.3390/ijerph18084283

**Published:** 2021-04-17

**Authors:** Olga Semenova, Julia Apalkova, Marina Butovskaya

**Affiliations:** Institute of Ethnology and Anthropology, Russian Academy of Sciences, 119334 Moscow, Russia; apalkova.julia@iea.ras.ru

**Keywords:** disease avoidance, sociality, spatial behavior, sex-specific strategy, pandemic, COVID-19, lockdown

## Abstract

Testing individual motivations for social activity in violation of the mandated lockdown regime is a challenging research topic for evolutionary psychology. To this purpose, we analyzed twenty popular weekly routes and the potential impact of sex and relationship status (single versus coupled) on the reported level of spatial-social activity during the quarantine in Russia between March and June 2020 (*N* = 492). Our study revealed a significant difference between men’s and women’s mobility: men, in general, tend to exhibit substantially higher spatial activity. The results have shown that individuals living on their own have more social interactions with friends and exhibit more profound spatial mobility via public transport. On the other hand, spatial activity of coupled individuals of both sexes were mostly devoted to solving a list of economic and matrimonial tasks. At the same time, men already cohabiting with a partner leave their homes for dating purposes more frequently than single men and women. We interpret these findings in the sense that both individual and sex-specific differences in observed sociality could be a result of a fine-tuned adaptive populational response to a contemporary virus threat, predominantly rooted in the evolution of behavioral strategies in the reproductive and economic spheres of each sex. Indeed, unlike women, coupled men have been preserving highly risky and intense social behavior during the COVID-19 pandemic.

## 1. Introduction

The emergence of profound advantages of collaboration within a social group drives sociality’s natural and social evolution [1]. Numerous fitness benefits of group living tend to increase with the number of group members in the society. Accordingly, a larger cluster of conspecifics could efficiently accumulate energetic resources, manifested as a direct biomass’s growth of all members and as infrastructures or stored resources and products [2]. The proposed advantages directly arise from communications between group members [3] where individuals connected via multiple associations with other members efficiently share signals and information [4].

However, social life presupposes the presence and consideration of both benefits and costs [5]. Well-known are the major drawbacks and high costs associated with the rapid transmission of pathogens and parasites in humans and other social animals living in groups through close and frequent contact [6,7,8]. The reproduction rate of directly transmitted pathogens is proportional to the availability of hosts and the frequency of their contacts [9,10]. Therefore, social animals (including humans) living in clusters of high local densities that interact closely with each other are particularly vulnerable to the threats posed by infectious diseases, especially so for emerging diseases such as the 1918 flu and COVID-19. As the risk and frequency of infectious diseases increases, this can impair a species’ ability to respond adequately, which may lead to high mortality and even jeopardizing the chances of survival of socially coherent groups.

Reducing the rate of spatial and social activity could be an important strategy for social animals to diminish the impact of directly transmitted diseases. The decline in the frequency of social interactions in response to a pathogen pressure may be obtained by changes in the behavior of infected and/or uninfected hosts. Infected hosts may decrease their sociality due to illness-related behaviors such as apathy and physical weakness. Clean hosts can reduce their risk of getting an infection by the adaptive response to cues of pathogen exposure within the entire population. Therefore, avoiding close contacts with conspecifics showing those cues of infection [11,12,13,14] or a general decrease in sociality when disease pressure became too high could be the main strategies of uninfected individuals.

Fine-tuned adaptive behavior assures taxa’s efficient response to the abruptly changing environmental settings. Spatial behavior and sociality play a crucial role in various biosocial processes, such as information flows, resource acquisition, reproduction, etc. [15,16,17,18]; these behavior patterns are substantially flexible and can be modified quickly in response to environmental cues. For example, behavioral responses such as dispersal or migration related to climate changes across all taxa were identified in 32.2% of 626 peer-reviewed papers as a common and successful adaptation [19]. Regardless of the nature of the threat, disease-induced reduction in spatial activity and sociality is likely to entail trade-offs and costs to all animals that are practicing social living in gropes, and humans are not an exception [20].

### 1.1. Modularity and Ingroup Sociality as an Adaptive Strategy

Assuming the taxa’s trade-offs connected to the losses sustained through sociality damages, a certain degree of modification in social structures might be anticipated (e.g., increased modularity). This transformation could potentially lower the costs of sociality for the entire group by implementing within-group contacts instead of outgroup interactions. [7,21]. For instance, in humans, recent investigation has shown that the circulation of COVID-19 sufficiently subsides in line with the growth of social group geographical and individual modularity within a studied social network of people [22]. In humans, such modularity may be represented by the growth of ingroup–outgroup distinctions [23,24]. Ingroup people treat outgroups as potential hosts of disease and this concern about pathogens drives considerable prejudice against outgroups. Group members could implicitly identify outgroup individuals as pathogen carriers at a high probability, especially those who have increased vulnerability to infection risks [25,26,27,28]. This modularity results in increased distance and avoidance of ‘foreign others’ in favor of well-known ingroup members of the societies [27,29,30,31].

A range of studies has demonstrated that a heightened disease threat can promote within-group sociality. For instance, cues of an illness can promote cohesion among the same group members, boost caregiving and helping behavior with trusted ingroup individuals [32] and enlarge family and kin connections [33].

Hence, sociality under the pathogen pressure connects to a certain degree of aversion to disparate foreigners belonging to outgroups. In contrast, sociality refers to the preferred association between similar individuals of the same group and even the same family. Within the society, expressing helping behavior could boost the individual prestige of such a caring member of the community and induce reciprocity further down the chain [33]. Group cohesion has many potential benefits for all its members. While losing access to the social network of allies, including friends and family, can be a source of psychological problems such as loneliness, anxiety, and depression [34]. Medical research states that social isolation could elevate the mortality risk from almost all human major disease. Similar harmful manifestation of losing sociality applies to all social animals and can be found in different taxa. Thus, induced stress, including social isolation and social adversity, may increase social disparities in health risk and mortality [35,36] through accelerating disease progression [37]. While ally social integration may promote more rapid improvement and recovery [38]. Therefore, it is supposed that even under the pathogen pressure a group cohesion has many potential benefits for all its members, as sociality may offset some of the pathogen’s risks caused by intense social life [39,40,41]. The last statement could also be valid for abundant groups of individuals and for high-density living areas in the case of humans. On the one hand it has been well documented that living in highly urbanized cities during a large-scale pandemic can be exceedingly devastating for human population, not only due to the high rate of pathogen transmission, but also because of the growing number of infected individuals strains, public health infrastructure and limited individuals’ access to adequate care [42,43]. On the other hand, ‘care’ implies a much broader phenomenon, including protection, nutrition, affiliative support, and psychological help. All of this might affect individual survival outcomes in a high-density areas and counterbalance the pathogen threat [44,45,46,47,48].

### 1.2. Negative Consequences of an Individual Social Isolation

In March 2020 in response to the novel challenges of COVID-19 pandemic’s threat the set of measures were offered by the World Health Organization to reduce the risk of infection spreading. In addition to hygienic recommendations such as frequent hand washing, disinfection of surfaces and objects, enhanced quarantine assures were implemented, which included the increased role for self-isolation and an abrupt reduction in social contacts. Thus, the new pathogen exposure forces us to distance ourselves from each other, dramatically lowering interpersonal sociality levels; as a result, many have truly begun to feel lonely.

Both the objective social adversity (such as the actual loss of social ties) and perceived social isolation and the loneliness feelings through various mechanisms, influence human health, affect morbidity and mortality rates [49,50,51], producing strong relationships between social factors, disease risk, and survival outcomes, similar to those associated with obesity [52]. It is supposed that a social group’s size and its quality have a profound effect on our susceptibility to various diseases and even death probability [53]. However, friendship is highly sensitive to rapidly lowering interpersonal interactions, leading to decay in perceived quality [54]. In addition to various psychological problems, the COVID-19 pandemic has forced people to limit or even totally cancel interactions with other people, consequently stimulating the unconscious need for physical contact, vision of other people’s faces, or the opportunity to spend time with friends: life becomes meaningless and converts into mere survival [55].

The dangerous new COVID-19 virus has swept the world at a time when an increasing number of people are living alone. The tendency to reduce the family to a single member became quite noticeable in the early twentieth century in industrialized countries, and this process accelerated in the 1960s. In 2019, 28% of U.S. households consisted of one person. Stockholm shows the maximum values of this trend towards a secluded lifestyle: for example, in 2012, 60% of households in Swedish cities consisted of only one person. In Russia, about 30% of households consist of one person, 30% of homes have two family members and 19% comprise three individuals. Four or more individuals constitute a family in Russia only in 19.4 cases out of 100 [56].

Given fact that a significant proportion of contemporary households have a reduced family size, and a growing amount of people nowadays live individually, our study raises an urgent question about the way how an individual lifestyle affects the individual sociality and spatial behavior as well as the readiness to comply with quarantine measures. 

The voluntary and mandated social distancing and lockdown quarantine procedures in human societies during the recent outbreak of COVID-19 created a unique opportunity to analyze the observed adjustment of human sociality in response to a novel pathogen exposure and compare these responses to theoretical predictions. Due to the COVID-19 pandemic, a self-isolation regime was announced on March 31 (RIA Rianovosti, Moscow, Russia, https://ria.ru/20200331/1569375432.html, accessed on 9 September 2020) and the quarantine regime was ultimately canceled only on 9 June 2020 (https://rg.ru/2020/03/05/moscow-ukaz12-reg-dok.html, accessed on 11 September 2020). During that period, we collected information about the persistent individual spatial activity in the Russian-language segment of the Internet. By doing so, we aimed to examine individual differences in weekly spatial activity between the singles and individuals living in pairs (cohabiting with a partner) in weekly outdoor activity during the lockdown regime.

### 1.3. Sex Differenses in Risk-Taking and Parental Investment and Respond for Novel Environmental Threats

Sex differences in propensity to risk-taking are universal [57] and may reflect Darwinian adaptations. For men, the willingness to take risks also related to reproductive benefits. According to evolutionary perspectives, sex differences in fertility, parental investment and divergent families’ roles set up greater challenges for men than for women in the intra-sexual competition for mating partners [58]. Higher testosterone levels in men potentially reduces fear, promoting risk taking behavior in males as a form of display which serves to intimidate same-sex rivals and advertise for individual’s quality as a sexual partner [59]. Women prefer men who demonstrate strength and the ability to acquire and maintain resources (which usually involves risk-taking), as these traits contributed to the success of ancestral reproduction [60]. Thus, the greater willingness of men to take risks, compared to women, in situations of sudden environmental changes and the emergence of new threats may be a psychological feature of men that has developed and consolidated in the process of evolution of the modern human species. Risk-taking influenced sexual selection over a long period of time could eventually become part of a male sexual strategy aimed at attracting and retaining female partners. For women, risk has always been an undesirable strategy, since along with the woman, her offspring (already existing and potential) are also at risk. Psychological studies have documented the greater fear in females which mediates women’s lower involvement in a range of risky activities A large number of animal and human studies have suggested that the prosocial and calm responses in females to a variety of environmental stressors are in conjunction with reproductive hormones and oxytocin mediated [61]. It is supposed that female reproductive success is enhanced by reduced mobility during reproductive periods, through lower energy expenditure and a reduction in accidents or predation [62,63]. In most mammals, including our species, maternal investment exceeds paternal investment. In our own species, fathers devote a much lower percentage of time to direct care than women do, and men’s interaction with their children is only about quarter of the maternal time of childcare [64,65]. Hence, females disproportionately bear the burden of parenting compared to males. In accordance with the fertility and parental care hypothesis, females have the higher costs attendant on mobility and related risk-taking activities, that advocate for the evolution of sex differences in spatial ability and mobility [66]. It is important to remember that the very preservation of the population depends on the pool of women who can take part in reproduction and then effectively ensure an infant’s survival outcomes; these biosocial premises provide an evolutionary platform for expecting this bias in risky behavior among two sexes.

### 1.4. Predictions and Hypothesis

As indicated by previous research, parasite-stress and virus infection correlate with the values of ingroup assortative sociality, assuming firm family ties [31]. It is well documented that the emergence of infectious diseases is positively associated with a strengthening of family ties; given that premise, we expected less sociality and outdoor social activity among individuals who reported cohabitation with a partner. On the other hand, behavioral responses to diseases are almost certainly costly for individuals living on their own and experiencing dramatic social isolation and low social integration, promoting stress. Therefore, individuals who live alone are expected to be less tolerant to the strict quarantine procedures and hence hypothetically could demonstrate more spatial and social activity in their weekly routine solving a variety of social and psychological tasks. In this study, we aimed to analyze a phenomenon of disease-mediated behavioral responses of two groups of individuals: those (1) living alone, and those (2) cohabiting with a partner. To answer the question, we examine the attendance of twenty popular routes by our respondents and estimate the potential impact of the sex of respondents on the range of the manifested spatial activity repertoire during the lockdown regime.

We also expect to detect a risk-sensitive response to a novel coronavirus general threat and a certain degree of variation in basic forms of social activity in our respondents. We hypothesize that the disease-avoidance responses are likely to be associated with respondents’ sex, and women who are the primary caregivers for the offspring are expected to be more avoidant and to demonstrate less outdoor activity than men, who are more preoccupied with resource acquisition tasks.

We assume that in any unstable and threatening situation, these evolutionarily based scenarios (risk avoidance for women and risk readiness for men) will be activated.

## 2. Materials and Methods

### 2.1. Sample and Data

The present study encompassed 490 participants (427 individuals from Russia; 21 individuals from Ukraine, 9 individuals from Kazakhstan, 8 from Belarus, 4 Germans, 2 Polish and Swiss each, and 19 nationalities). The mean age of the participants was 35.49 (SD = 10.03) and the sample was gender-balanced (female *N* = 225; men *N* = 265) and represents heterosexual adult individuals (minimum 17 y, mean age is 35 y).

### 2.2. Procedures

The invitation of participants for the study was made largely on Russian-language social media and entertainment websites. A call was made to potential participants to take part in an online survey about their spatial activity and mobility during the corona virus pandemic. A secure data collection software providing unique subject IDs not allowing for individual identification was used and the participants had to give informed consent to the use of data provided by them. A compensation for taking part in the survey was not offered.

The study was conducted from 29 March 2020 to 27 June 2020 (Median 12 April 2020), a period of time in Russia during which the corona pandemic was fought with strict quarantine regulations.

### 2.3. Field Measurements

This study used a self-report online survey, elaborated by the research group of IEA RAS, Moscow, Russia, and aimed to fixate the pandemic mobility patterns (COVID-19 PMP questionary). This online survey was firstly adopted and introduced by authors in March 2020 to measure mobility and spatial behavior during the COVID-19 pandemic. Emphasis was made on individual spatial behavior and on the frequency of movement away from home during the previous week. We also asked about the frequency of visits to 20 popular destinations in an individual weekly activity. Additionally, respondents were provided with an opportunity to mention extra destinations, not listed by us, under the name of “other”. All participants were asked to provide information on their key biosocial attributes such as age, sex and marital status. Additionally, we asked if respondents cohabited with his/her partner within one household.

### 2.4. Statistical Analysis

Statistical analysis was performed using SPSS statistical software (IBM Corp, Armonk, NY, USA). Among others we performed a generalized linear model (GLM) analyses and *t*-test statistics.

## 3. Results

In this study, we analyzed the twenty most popular urban destinations in weekly activity outside of home ranged from 0 to 8 points (0 points—never, 1 point—ones a week, 2 points—twice a week…, 7 points—daily, 8 points—more than ones in a day). It was found that most popular destination was the groceries with the mean frequencies approximately 3.5 times a week. The work places gained the second position with a mean frequency of 2.7 times a week (Table 1).

We also analyzed the potential difference in frequency of activities outside of home depending on the whole spectrum of the examined goals of social activity across single and coupled individuals. We performed a set of *t*-tests for outdoor activity for two groups of individuals (coupled and singleton) which revealed statistically significant differences in the particular types of social activities. Individuals who reported cohabitation with a partner were more active in visiting: (1) pharmacies (*t* = 2.580, *Df* = 396, *p*-value = 0.01); (2) garages and auto shops (*t* = 2.150, *Df* = 378, *p*-value = 0.032); (3) childcare facilities (*t* = 3.055, *Df* = 379, *p*-value = 0.002); (4) outdoor activities with kids (*t* = 3.603, *Df* = 396, *p*-value < 0.001); (5) meeting with sexual partners (dating) (*t* = 3.612, *Df* = 381, *p*-value < 0.001); (6) walking pets (*t* = 2.037, *Df* = 386, *p*-value = 0.042). Meanwhile, single individuals more often (1) visited their friends (*t* = −2.576, *Df* = 387, *p*-value = 0.01) and (2) used public transport (*t* = −2.454, *Df* = 394, *p*-value = 0.015).

Figure 1 demonstrates pie/donut charts representing the important weekly activities. The weekly outdoor activity of singletons is presented in grey color and coupled individuals’ activity is presented in green.

We performed a two-factor analysis of variance to reveal the investment of sex (men versus women) and cohabitation with a partner (singleton versus coupled individual) factors on socially manifested outdoor spatial activity (Figure 2).

General levels of outdoor spatial activity were significantly higher for men cohabiting with a partner (*R*-square = 0.065; *F*(sex) = 11.761; *Df* = 1; *p*-value < 0.01; men *N* = 225; women *N* = 263). Interaction of both studied factors was also significant (cohabitation * sex, *F* =16.424, *Df* = 1, *p*-value < 0.001), which means that for men and women the factor of cohabiting with a partner affected their level of outdoor spatial activity differently.

In the next stage, we analyzed potential differences in the frequency of social activities depending on the activity aims across men and women. We performed a set of *t*-tests for males’ versus females’ outdoor activities frequencies. The analyses revealed the statistical difference in the four types of social activities where men demonstrated a higher level of outdoor spatial activities: (1) work (*t* = 2.767, *Df* = 414, *p*-value = 0.006); (2) meeting friends (*t* = 2.216, *Df* = 385, *p*-value = 0.027); (3) and meeting with sexual partner (dating) (*t* = 4.115, *Df* = 380, *p*-value < 0.001). The only outdoor activity where women demonstrated statistically significantly higher spatial mobility was walking outdoors with pets (*t* = −2.065, *Df* = 384, *p*-value = 0.04). Figure 3 demonstrates the pie/donut charts representing the important weekly activities.

Based on the revealed differences among men and women in reported frequencies in several domains of outdoor spatial activities, we performed four sets of a two-factor analysis of variance tests where we additionally examined the influence of the factor of cohabitation with a partner (singleton versus coupled individual) on men and women (Figure 4). Analysis of variance revealed that men cohabitating with their partners visit work significantly more frequently during a week (*R*-square = 0.017, *F* = 6.341, *p*-value = 0.012, *Df* = 1). Single individuals significantly more frequently meet their friends during a week in comparison to coupled individuals (*R*-square = 0.018, *F* = 4.912, *p*-value = 0.027, *Df* = 1), also men tend to demonstrate higher sociality in friendship than women (*F* = 4.318, *p*-value = 0.038, *Df* = 1). Men already cohabiting with a partner left their homes to meet with their current or potential sexual partners significantly more frequently during a week in comparison to both groups: singletons individuals (*R*-square = 0.09, *F* = 16.117, *p*-value < 0.001, *Df* = 1), and women (*F* = 16.313, *p*-value < 0.001, *Df* = 1), interactions (*F* = 6.261, *p*-value = 0.013, *Df* = 1). Individuals cohabiting with a partner walked their pets significantly more frequently during a week in comparison to single individuals (*R*-square = 0.016, *F* = 4.3, *p*-value < 0.039, *Df* = 1); however, women walk pets more often than men (*F*= 3.107, *p*-value < 0.079, *Df* = 1), interactions of both factors were not significant (*F* = 0.69, *p*-value = 0.43, *Df* = 1). This result infers to spending time walking a pet and does not refer to the fact of having a pet to walk, as we did not control for its presence in a household.

## 4. Discussion

The analyses of a weekly routine spatial activity and the individual level of social contacts during the mandated and (or) voluntary lockdown in March–June 2020 in Russia revealed the key biosocial differences between the sexes: men were significantly much more active during quarantine and demonstrated higher regularity in establishing personal social contacts than women.

Theory predicts that disease-mediated reductions in sociality within both sexes potentially are associated with trade-offs and greater costs, including elevating energetic and resource output, lost essential opportunities, and other psychologically adverse outcomes. Implementing flexible and fine-tuned external social roles for individuals of different sexes, where men, being a primary provider of a family, used to perform risky social contacts, allows women to reduce their degree of spatial and social activity significantly. 

Our data have shown that coupled men who live with a partner significantly more often leave their place of residence compared to single men (Figure 2). The opposite results were obtained for women. Single women demonstrate more spatial activity and a higher level of risky social contacts during the lockdown within a one-week time interval than women living with a partner. It is important to note that the sexual division of family roles in mobility is positively associated with the resource provisioning tasks and work attendance as men who are cohabiting with a partner reported more frequent spatial activities attributed to work duties; however, for singles, the outdoor work activity was irrelevant to the sex of the respondent (Figure 4A).

This sociospatial gender specialization could potentially serve their common reproductive interests and childrearing tasks. Data from other anthropological studies provides evidence for less direct childcare of men compared to female’s direct effort, and that discrepancy among family roles is more profound in agrarian and pastoral societies, where men have the lowest level of proximity to infants [64]. While males generally help little with children, they are important economic contributors in many traditional societies, where they provide the majority of calories and most of the protein into the family diet [67,68]. According to the movement ecology theory, this sexual division of labor and sex differences in spatial behavior and landscape use likely had evolved in human hunter-gatherer’s past as a specialization of each sex either on the targeting relatively rare and moving game or landscape food searching tasks [60,63,69]. Recent anthropological research in traditional societies has supported this theoretical concept of gender specialization. In the hunter-gatherer context of human evolution, the distribution of the targeted game and resources density could have resulted in the development of prominent gender specialization and differentiation in spatial and social behavior, where males exhibit greater mobility accompanied by more risky and explorative spatial activity patterns than women, as the former used to seek rarer and more mobile resources [60,70,71].

We suppose that this gender-roles stratification in spatial activity and sociality may still be adaptive during pandemics in the modern world. Particularly, the basic principles of sexual division of labor may explain our findings on socioeconomic activity division within a couples.

The costs of social distancing in humans are dramatic. Both induce social isolation constituting an actual loss of social integration and subjective feeling of social isolation, including the feeling of loneliness and perceived lack of social engagement with allies and friends could induced stress and cause a substantial elevation in depression rates, essential decreasing the quality of life, and could even raise the mortality rate through the subsiding an immune system, and increased disease susceptibility [72,73,74]. Regarding rising costs for individuals who reported being single and at a high risk of living alone during the lockdown, we expected to receive evidence of a flexible response (adaptation) to that mandated social isolation. For instance, we expected to observe a relatively higher level of external social contacts among singletons and individuals without a partner. Indeed, for single individuals, we detected significantly higher outdoor activity levels and an increase in interaction with friends for both sexes, compared with respondents living in pairs; single individuals reported more frequent spatial behavior related to public transport usage. A relatively higher level of spatial activity and supporting offline social connections in single individuals is likely a mechanism that works as a buffer against depression and loneliness. Simultaneously, coupled individuals of both sexes were much more active in solving a list of purely economic and general matrimonial tasks, which included visiting pharmacies, garage workshops, childcare facilities, walking pets, etc. Hence, we revealed the two principally different sets of activities in singletons and coupled individuals during the lockdown. While singles tended to seek psychological relief, coupled individuals were solving the family’s basic economic and recreative tasks.

Results indicate that interacting with pets and other animals, including outdoor activities, was not higher among single individuals. However, data showed that women more often performed outdoor trips with pets during the quarantine. These findings can partly support the hypothesis of less risky behavior among women. They significantly more often tend to reduce contact with conspecifics while performing necessary spatial activities with pets.

Disease-mediated reductions in social interactions can pose a particular challenge for social scientists as reducing sociosexual contacts leads to subsequential loss of populational fecundity. Our findings are in line with the theory of sex-differentiated reproduction strategies [75,76,77] and indicate that men have been preserving a higher level of sociosexual contacts during the current pandemic of COVID-19 compared to women.

It is important to mention that our data showed that sexual activity was significantly higher among men who already have a partner within their household. Coupled men reported substantially higher outdoor activity for extramarital mating efforts and courting more often than women and single men during the lockdown period. This result was puzzling as the evolution of risk-sensitive responses for men who already have a sexual partner should favor distancing and decrease their mating efforts. For instance, Wilke with coauthors [78] had shown that, in the domains of mate retention and attraction, single individuals gave higher scores on propensity of risk-taking behavior than those who were married or in a committed relationship. Contradicting to the Wilke with coauthors study, our results have shown that coupled men demonstrated an even higher level of sociosexual activity than single men. This finding could not be attributed to the respondents’ age. We did not find any correlation with biological age in the supplementary GLM models examining interactions of sex, marital status, and sexual activity. We suppose that this surprising finding of a relatively higher level of sexual activity in the cohort of coupled men is in line with the hypotheses of disease-mediated response and sex-differentiated reproduction strategies. As the parasite-stress or the ecological pressure of a new infectious disease (EIDs) has dramatically risen, the upholding of an appropriate amount of mating and reproductive efforts in the entire population had become a vital task. During the emergence of a novel disease, the importance of reproducing immune compensated and healthy offspring simultaneously increases.

It is important to note that the data show that the level of male sexual activity was negatively associated with the anxiety level attributed to the virus threat (*R*-squared = 0.022; b= −0.16; *F* = 8.68; *p*-value = 0.003). The observed correlation argued for a specific psychological mechanism that potentially lowers the feeling of danger and enhances the risky behavior in sexually active men. We conclude that in terms of behavioral response within a sex domain, successful men have been preserving highly risky and intense social behavior regardless of quarantine restriction on social contacts. Future studies could investigate this prediction and analyze female sexual preferences in terms of growing females’ interest in masculine men who were already successful in gaining a mate. Female’s demand in a sexual variety in order to gain a genetic diversity for their children when pathogens abound could also be tested.

Other potential explanations for the observed difference in sociosexual activity could be associated with the variety of biosocial and psychological selection factors influencing individual behavior patterns during the present pandemic. For instance, single individuals could be suffering higher levels of stress, and therefore males living on their own could be more socially and sexually disabled by anxiety.

Individual response to a pathogen stress could be also seen through the lens of a life history theory framework (LH), which describes variation in the allocation of finite bioenergetic resources among somatic, mating and parental tasks. Psychological studies evident that the fast life history strategy has been linked to poorer health outcomes and more risker behavior [79]. More active sexual and spatial activities in some males could advocate for the presence of fast LH traits. Males who perform a fast LH strategy could pursue riskier, more competitive interpersonal relationships that strain social cooperation. They can have an incentive to maximize their own personal reproductive success and ‘free ride’ on the cooperation of others by following individual sexual interest [80].

### Future Agenda

We suppose that including cross-cultural data in a future study of the observed violation of the mandated lockdown regime will facilitate our understanding of the efficiency of various public health policies, guidelines, aiming to promote a social distancing. As we know, the most extreme version of such measures involved strict ‘lockdowns’ which profoundly limited our social activity, was almost synchronously implemented across the globe. From the inclusive fitness theory’s evolutionary perspective, keeping spatial and social activity during the disease outbreak could be defined as harmful for the public health goals and social cooperation tasks. However, a certain degree of sociality could be seen as adaptive incentives for the survival outcomes on the populational level. There-fore, a broader study design is needed to understand and assess the possible negative or positive long-term effects of the various disease-mediated individual responses on the pandemic COVID-19. At the same time, the authors did not pursue that goal in this particular study. Utilizing a collected during COVID-19 first wave exposure dataset, we had mainly implied to fix and analyze the hypothetical preexisting patterns of the social and spatial response in the studied population on the new pathogen pressure, assuming that in our species as well as any other highly social animals, could have a developed adaptive response mechanism on the recurrent pathogen invasion. Examining the key influential factors such as biological sex and relationship status allowed us to define a possible perspective direction for future studies. At the same time, we aim to enlarge the sample and collect the data in a post lockdown environment to compare the patterns of spatial behavior. Utilizing a bigger sample size permits us in the future to test more distinct and definitive predictions such as the potential effect of children on social behaviour, individual socioeconomic status and cultural context. In addition, the inclusion of partner choice in the study design could help to elaborate and develop understanding of the greater willingness of men to take risks compared to women during the sudden environmental changes. We assume that comparative analysis of the sexual behavior of people during lockdown with that after COVID-19 is needed.

## 5. Conclusions

Our analysis has shown a profound gender-roles stratification in spatial and social activity during the mandated and (or) voluntary lockdown in March–June 2020 in Russia, which we suppose may serve an adaptive purpose for the entire population facing a new pathogen invasion. The sexual division of family roles among coupled individuals may explain our findings on socioeconomic activity difference among men and women. In a context of new pathogen pressure, this sex-role differentiation could potentially decrease disease transmission among all family members (including children) and hence among the entire population. This sociosexual specialization facilitates the reduction of the overwhelming cost of essential sociality during the pandemic. We suggest that this adaptation could minimize the pathogen burned by maintaining growth in predominantly within-group or even within-family social interactions, promoting group cohesion, caregiving, and helping behavior within the family units and units of social allies.

At the same time, an elevated level of spatial activity and grown offline social connections with friends in single individuals are likely could be a mechanism that works as a buffer against depression and loneliness among individuals living on their own during several months in social isolation in accordance with public health guidelines, aiming to promote a social distancing during the first wave of COVID-19 outbreak. 

In contrast, a higher level of spatial activity associated with the growing intensity of mating effort in men supports the hypotheses of disease-mediated behavioral response to the new virus invasion implying strengthening gender differentiation in reproductive strategies. Our data positively demonstrated that coupled men courted more often than women and single men, the former performed substantially higher outdoor activity for extramarital mating purposes. This finding indicated that men who were already successful in gaining a mate have been preserving highly risky and intense sexual behavior regardless of quarantine restriction on social contacts. We suppose that for the entire population, which faced a new infectious disease (EIDs), an observed behavioral response in a cohort of coupled men could serve the goal of upholding an appropriate amount of mating and reproductive activity, as the importance of reproducing immune compensated and healthy offspring had become a vital task. Therefore, from an evolutionary perspective, there are no premises for a sufficient lowering of the amount of sociosexual activity among men who already demonstrated a certain degree of success within the intrasexual competition.

## Figures and Tables

**Figure 1 ijerph-18-04283-f001:**
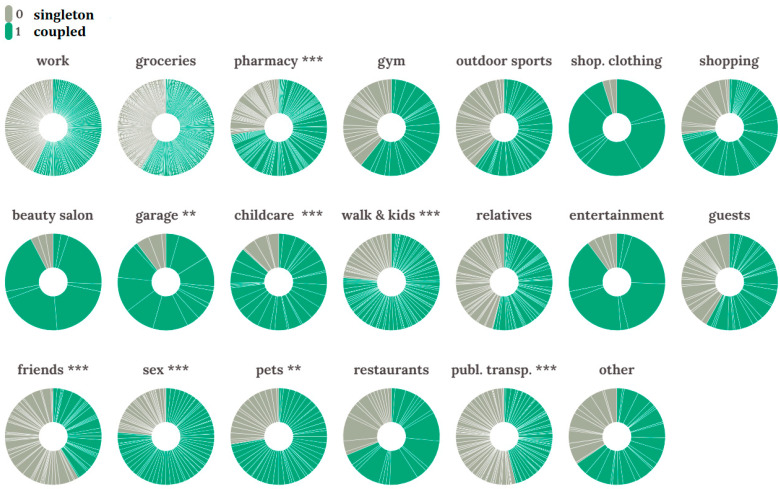
Pie/donut charts representing important weekly activities. Grey color—singletons outdoor activities; green color represents activities of coupled individuals within each destination (** represents a statistically significant difference with *p*-value < 0.05, *** represents statistically significant difference with *p*-value < 0.001).

**Figure 2 ijerph-18-04283-f002:**
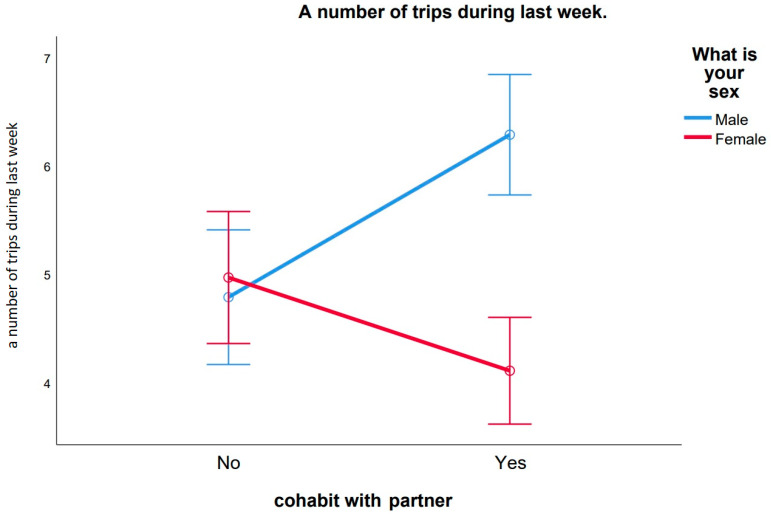
An average number of outdoor trips with 95% confidence interval (CI) for men and women in one week during quarantine in Russia between March and June 2020, depending on their relationship status (single versus coupled).

**Figure 3 ijerph-18-04283-f003:**
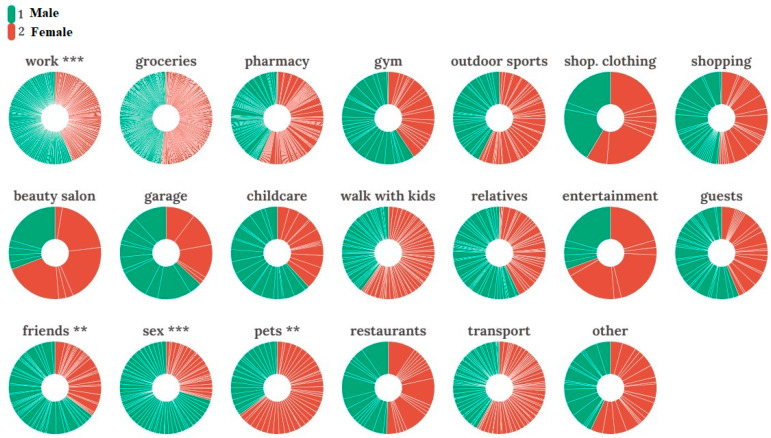
Pie/donut charts representing important weekly activities. Red color—women’s part of activities; green color represents male’s activities within each destination (** represents a statistically significant difference between sexes with *p*-value < 0.05; *** represents statistically significant difference with *p*-value < 0.001).

**Figure 4 ijerph-18-04283-f004:**
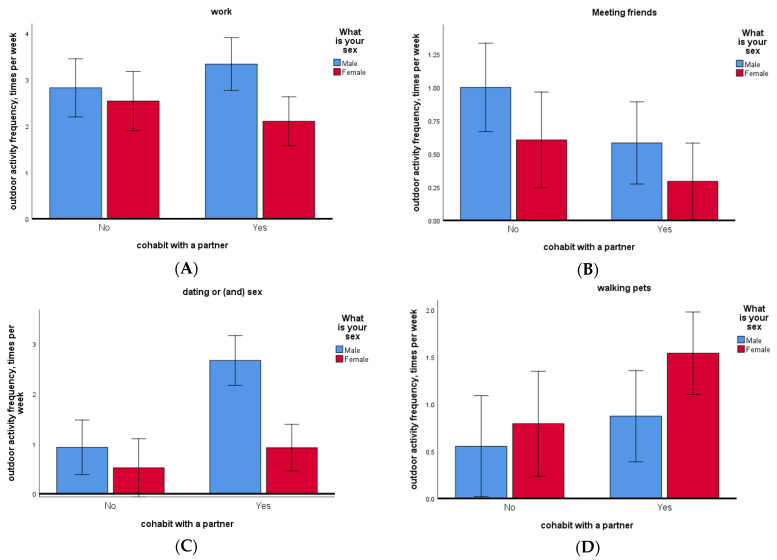
The frequency of outdoor spatial activity associated with four types of destinations (**A**) work, (**B**) friends, (**C**) meeting sexual partners, dating, (**D**) walking pets. The data are presented for cohabiting with a partner or single individuals. Men are presented in blue, and women in red.

**Table 1 ijerph-18-04283-t001:** The descriptive statistic of the frequency of visits to 20 potential destinations by respondents.

Destinations	*N*	Min	Max	Mean	SD
Work	418	0	8	2.70	3.072
2. Groceries	464	0	8	3.47	2.371
3. Pharmacy	398	0	8	0.86	1.653
4. Gym	381	0	8	0.52	1.655
5. Outdoor sports	388	0	8	0.73	1.897
6. Shopping (clothes)	381	0	8	0.13	0.909
7. Shopping (home and garden)	385	0	8	0.38	1.351
8. Beauty salon	378	0	8	0.10	0.829
9. Vehicle workshops (garages)	380	0	8	0.18	1.027
10. Children entatainment and facilities	381	0	8	0.50	1.806
11. Walk outdoors with kids	398	0	8	1.19	2.390
12. Assistance relatives	246	0	8	1.20	2.143
13. Entertainment, clubs, bars	377	0	8	0.10	0.807
14. Guests (come out)	384	0	8	0.44	1.409
15. Meeting friends	389	0	8	0.61	1.638
16. Meeting sexual partner (dating)	383	0	8	1.31	2.691
17. Walking pets	388	0	8	1.02	2.514
18. Restaurants, cafe	380	0	8	0.22	0.981
19. Public transport	396	0	8	1.26	2.448
20. Other	256	0	8	0.64	1.933
*N*	160				

## Data Availability

Data and visualization are available online: https://app.flourish.studio/visualisation/4743864/ accessed on 15 April 2021.

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
