# Peer review of "Spatial and Social Behavior of Single and Coupled Individuals of Both Sexes during COVID-19 Lockdown Regime in Russia"

_ijerph, 2021, doi:10.3390/ijerph18084283_

Round 1

Reviewer 1 Report

This study examined social behaviours of people during COVID-19 lockdown, and found that they were associated with sex (men/women) and relationship status (single/coupled). The authors discussed their findings from evolutionary perspectives (sexual division/risk preference).

I think that the results would be interesting and valuable empirical evidence of human behavior in the novel socioecological environment. However, their relevance to evolutionary theory is not clear for me and seemed to be just speculation.

Below I leave some comments that might help to further revise the manuscript.

(1) Lines 168-170: … women who are the primary caregivers for the offspring

In human child-rearing, parental care by both mother and father will be crucial, so this scenario should be considered more carefully. Could the authors explain the sociocultural context of childcare in the study region? I also wonder that why the authors did not directly analyse the effect of existing/potential children on social behaviour to test this prediction.

(2) Lines 179-187: … greater willingness of men to take risks, compared to women

This might be the case, but the present study did not investigate the process of partner choice sufficiently. Therefore, I think that this prediction was not tested well.

(3) There have been some papers that discusses human behaviour under COVID-related situations from evolutionary perspectives. The following two articles might be helpful to reframe the introduction section of the present paper. Please note that I do NOT compel the authors to cite them in your manuscript (and I am not the author of them). I would appreciate your consideration.

https://doi.org/10.1002/ajhb.23485

https://doi.org/10.1093/emph/eoaa038

(4) Figure 2

What do error bars indicate? Please add information in the figure legend. The title of y-axis will be needed too.

(5) Could the authors compare social behaviour of people during lockdown with that before COVID-19? Also, could one's socioeconomic status be associated with social behaviour? Adding some information or discussion would be appreciated.

(6) Lines 304-317

Could the authors add more detailed explanation and references? This is overdiscussion as it is.

(7) Lines 327-332

Could the authors add references that support these pathways? The present analysis did not show any causal relationships between factors, so additional information will be needed.

(8) Lines 207-208: This study used a preregistered self-report questionnaire

Was the questionnaire pre-registered in this journal? If so, I think that the same referee(s) should review the present manuscript again. (Cc: the handling editor)

(9) Lines 390-391: The study protocol was approved by ...

If there is approval number, please show it.

(10) Could the authors add data availability statement between the end of the discussion section and the reference list?

(11) Line 219: GLM -> generalized linear model (GLM)

Author Response

(1) Lines 168-170: … women who are the primary caregivers for the offspring

In human child-rearing, parental care by both mother and father will be crucial, so this scenario should be considered more carefully.

               We have enlarged this paragraph with more detailed theoretical and                   empirical incentives and examples. Lines 173-179

 Could the authors explain the sociocultural context of childcare in the study region?

                  In accordance with the study's chosen design, which focused on the two important biosocial attributes (sex and single/couple status) and searching for universal adaptation mechanisms to abrupt socio-ecological changes, we did not investigate the cultural specifics of particular behavioral patterns. 

                   That's why we didn't include describing childcare's socio-cultural context in the theoretical part of the study. We mention childcare only in the context of women's risk aversion as a universal form of behavioral adaptation to the pathogen threat that guarantees the population better survival by limiting spatial -social contact of women of reproductive age as well as their offspring.

                 We suppose that this could be a future agenda aim as the cross-                          cultural analyses can better reveal the Russian socio-cultural context.

I also wonder that why the authors did not directly analyse the effect of existing/potential children on social behaviour to test this prediction.

                       The evolution of sex differences in spatial ability could be explain in a frame of the fertility and parental care hypothesis, which states that female reproductive success is enhanced by reduced mobility during all reproductive periods, through lower energy expenditure and a reduction in accidents or predation. Under this theoretical concept, we suppose that mobility differences between men and women could be tested independently from the actual presence of children.

                      Also, we have to note that our sample size was not big enough. It is only about five hundred participants; therefore, we cannot include more predictors in the model for statistical reasons. As we already examined 20 routes destinations and two exogenous factors (single-versus couple state and two sexes: males versus females). It has been recommended to have each fraction of subsample minimum 20 participants to ensure the results' relevance, and more predictors would split the sample into smaller groups than required.

                     Given that premises,  we had to generalize the results and the theoretical design of the study. However, we think this is one of the perspective ideas for future studies.

(2) Lines 179-187: … greater willingness of men to take risks, compared to women

“Sex differences in propensity to risk-taking are universal [57] and may reflect Darwinian adaptations. For men, the willingness to take risks also related to reproductive benefits. According to evolutionary perspectives, sex differences in fertility, parental contributions and division families’ roles set up greater challenges for men than for women in the intra-sexual competition for mating partners [58]. Women prefer men who demonstrate strength and the ability to acquire and maintain resources (which usually involves risk-taking), as these traits contributed to the success of ancestral reproduction [59].  Thus, the greater willingness of men to take risks, compared to women, in situations of sudden environmental changes and the emergence of new threats may be a psychological feature of men that has developed and consolidated in the process of evolution of the modern human species. Risk-taking influenced sexual selection over a long period of time could eventually become part of a male sexual strategy aimed at attracting and retaining female partners”

This might be the case, but the present study did not investigate the process of partner choice sufficiently. Therefore, I think that this prediction was not tested well.

                   As we have said before, due to the limited sample size, we had to generalize our study and design to a certain extent. And in the current study, we did not investigate the potential group of factors that could increase ( or mitigate) the dangerous environment during the mating searching process and sexual intercourse itself, assuming that any extramarital interiors and even meeting a new sexual partner always denote a risky activity during a pandemic.

               At the same time, we do not agree that this prediction has not been tested well. We based a hypothesis of the unified assessment of any meeting as dangerous behavior on the theoretical premises introduced at the beginning of the present manuscript. According to them – any close social contact could be regarded as hazardous in the pandemic environment.

(3) There have been some papers that discusses human behaviour under COVID-related situations from evolutionary perspectives. The following two articles might be helpful to reframe the introduction section of the present paper. Please note that I do NOT compel the authors to cite them in your manuscript (and I am not the author of them). I would appreciate your consideration.

https://doi.org/10.1002/ajhb.23485

https://doi.org/10.1093/emph/eoaa038

                We agree with most of the theoretical statements in the given recommended papers, especially concerning a reproduction issue. And we had included into discussion section the Life history theoretical premises influencing the individual patterns of socio-sexual risky behavior during the lockdown.

               At the same time, we suppose that some theoretical discrepancies between our manuscript and the two recommended papers bring us back to the well-known dilemma of kin selection versus group selection in explaining sociality. As can be shown, neither need be selected over the other(Bahar, 2017). Lines 441-450

                 In this current paper, we had no intention to go too far in the theoretical issues of the sociality and multilevel selection concepts ( Hamilton 1975). Our article had the incentives to examine the observed patterns of spatial behavior utilizing the hypothesized adaptations on the populational level. Plans of future research in human spatial behavior could include a more precise design examining the cooperation and sociality issues during the pandemic threat.

(4) Figure 2

What do error bars indicate? Please add information in the figure legend. The title of y-axis will be needed too.

            Thank you for this comment. We have added the CI and y-axis                               descriptions

(5) Could the authors compare social behaviour of people during lockdown with that before COVID-19?.

                  Unfortunately, no, as we have not used this survey before March                      2020. Therefore, this comparative analysis would be possible to                                 conduct  in the future. We can enlarge the current sample later and                     compare the impact of lockdown and post lockdown spatial                                  behavior.

  Also, could one's socioeconomic status be associated with social behaviour? Adding some information or discussion would be appreciated

                  The impact of economic issues and the individual variation in anxiety                 had been investigated in the additional paper. Here we focused only                       on fixing the observed behavior patterns without including                                    additional factors due to the sample size statistical limitations.

(6) Lines 304-317

“Implementing flexible and fine-tuned external social roles for individuals of different sexes, where men are a primary provider of a family performing risky social contacts, allows women to significantly reduce their degree of spatial and social activity and focus on providing care for the common offspring. This sexual division of labor and sex differences in spatial behavior and landscape use is likely to have evolved in human hunter-gatherer's past during the process of targeted game and searching food according to the theory in movement ecology [60]”

Could the authors add more detailed explanation and references? This is overdiscussion as it is.

                    We have added more explanations, lines 331-350

(7) Lines 327-332

“The costs of social distancing in humans are dramatic. Both induced social isolation constituting an actual loss of social integration and subjective feeling of social isolation, including the feeling of loneliness and perceived lack of social engagement with allies and friends could induced stress and cause a substantial elevation in depression rates, essential decreasing the quality of life, and could even raise the mortality rate through the subsiding an immune system, and increased disease susceptibility”

Could the authors add references that support these pathways? The present analysis did not show any causal relationships between factors, so additional information will be needed.

                    We have added references, line 375

(8) Lines 207-208: This study used a preregistered self-report questionnaire

Was the questionnaire pre-registered in this journal? If so, I think that the same referee(s) should review

                    We are sorry, we are not native-speaking authors, and we misused                       this term.

                  COVID-19 lockdown was an extraordinary situation, and existing in                     the literature, surveys about spatial behavior were not fitted well into                     the new lockdown environment. The current survey was offered and                    first introduced by our team. And after the Institutional ethical                               committee's approval, it was urgently implemented.

                   Lines 234-237

(9) Lines 390-391: The study protocol was approved by ...

Institutional Review board of IEA RAS March 19, 2021

If there is approval number, please show it.

                        There is no approval number.

(10) Could the authors add data availability statement between the end of the discussion section and the reference list?

                     We added the data availability

(11) Line 219: GLM -> generalized linear model (GLM)

Reviewer 2 Report

This is a highly original and well executed study. It is theoretically sound and cites the relevant literature. It includes some creative interpretations of significant findings. It raises important social policy issues by implication. For example, at least in the US, the fact that under Covid more women than men have become unemployed is cited as a great injustice to women. However, these data suggest a benefit: less infection risk.. Also, in Russia it seems that couples women work more than single ones. This might be difference elsewhere, and points to the value of cross-cultural researches such as this one. The data on coupled men courting more than single men are really fascinating and might be discussed further. Do women prefer sexual variety more in a pandemic because of a gain in genetic diversity for their children where pathogens abound? Would disease adversity make them more interested in masculine men who were already successful in gaining a mate? The evolutionary perspective, as exemplified by this study, raises neglected questions about family structure and sex differences. It is certainly topical given the pandemic. I would suggest adding more evolutionary analysis, such as health implications for children of coupled or single parents. 

The writing is very good for a non-native English speaker. However, the writing could be more idiomatic. Let me pose some specific suggestions for writing mechanics and some other matters:

Line 42--might be better not to refer to the Spanish flu, which unfairly stigmatized that country for its journalistic openness.. How about 1918 flu?

57f--use such as or etc., not both

85f--incomplete sentence

86f--sociality may not refer only to positive social contacts, although admittedly the word comes from socius, L. ally

93--this sentence is confusing and contains two misspellings

96--awkward sentence...and accelerate?

100--needs topic sentence that starts and defines the paragraph..."Losing" is unnecessary

114f--awkward, run-on sentence (too long)

125--new paragraph

136--delete the

145--on, not On

154--semicolon after ties...we expected

157--own and

158--tolerant of...expected, not supposed

168--be associated with

175--divergent family roles

176f--men take risks in fighting each other for mates too...might mention higher testosterone in men reduces fear...this is true of other male animals too...might cite K Hawkes' show-off hypothesis here...males'  risk taking intimidates rivals as well as impressing females

186--cite Ann Campbell on females' reluctance to fight and jeopardize their young...also cite S E Taylor in Psych Review on oxytocin secretion in threatened female animals--the tend-and-befriend response

191--any exclusions for sex orientation?

211--delete most

225--work places

230--performed

232--delete their

251-delete A...omit blue and red lines

259--performed

261--a higher

263--and meeting

285ff--clarify that this does not refer to rate of walking one's pet or having a pet to walk, but, rather, to spending time walking a pet

299--are not to

330--induce, not induced

338--with not for

347--family's

348--other animals

357--preserving a higher level

361--extramarital mating effort?

362--reproductive

378--recast as individual selection...e.g., single men may be disabled by anxiety

382--show

383--was negatively

384--argued not evidenced

Figure 4--are single vs. coupled labeled correctly? 

Author Response

The data on coupled men courting more than single men are really fascinating and might be discussed further. Do women prefer sexual variety more in a pandemic because of a gain in genetic diversity for their children where pathogens abound? Would disease adversity make them more interested in masculine men who were already successful in gaining a mate?

           Thank you for this comments.  We have added this issue into the possible future goals

191--any exclusions for sex orientation?

‘Risk-taking influenced sexual selection over a long period of time could eventually become part of a male sexual strategy aimed at attracting and retaining female partners. For women, risk has always been an undesirable strategy, since along with the woman, her offspring (already existing and potential) are also at risk.’

           We have data on heterosexuals only.  We may consider collecting data                  from non-heterosexuals, and then we can make a stronger statement.                   But heterosexuals are (still) the majority in terms of frequency of sexual               orientation, and we see little reason to speculate about ~7%                                homosexuals -- that is another study topic. In our investigation, we aim                    to  revile mechanisms of universal behavior for the majority of the                     population in the conditions of abrupt socio-ecological changes.

285ff--clarify that this does not refer to rate of walking one's pet or having a pet to walk, but, rather, to spending time walking a pet

                 We have added this clarification

378--recast as individual selection...e.g., single men may be disabled by anxiety

                  We have added this prediction into discussion

Figure 4--are single vs. coupled labeled correctly?

            Yes, they are correct.

Round 2

Reviewer 1 Report

Thank you very much for the revision. You dealt with my concern carefully. I think that the current manuscript will be acceptable for publication.